# Some Like It Hot: Investigating Thermoregulatory Behavior of Carcharhinid Sharks in a Natural Environment with Artificially Elevated Temperatures

Adi Barash [1,2,*], Aviad Scheinin [1,3], Eyal Bigal [1,3], Ziv Zemah Shamir [1,3], Stephane Martinez [1,3], Aileen Davidi [2], Yotam Fadida [4], Renanel Pickholtz [5] and Dan Tchernov [1,3]

1 Leon Charney School of Marine Sciences, University of Haifa, Haifa 3498838, Israel
2 Sharks in Israel, NGO for the Conservation of Sharks and Rays in Israel and the Mediterranean Sea, Kibutz Amir, Upper Galilee 1214000, Israel
3 Morris Kahn Marine Research Station, Sdot-Yam, Haifa 3498838, Israel
4 Israel Oceanographic and Limnological Research, Haifa 3108000, Israel
5 School of Zoology, Faculty of Life Science, Tel Aviv University, Tel Aviv 6997801, Israel
* Correspondence: adibarash@hotmail.com

**Abstract:** Global warming raises seawater temperatures and creates changes which have been found to affect the movement of large migrating marine species. Understanding the thermal niches of marine species could prove essential to anticipate how the future climate will alter migrations, and how conservation efforts will have to change accordingly. Orot Rabin power station in Hadera, Israel uses seawater to cool its turbine and releases the warm water back into the Mediterranean Sea. As a result, a marine area with artificially elevated temperatures is created around the effluent. Every winter in the past two decades, this area attracts sharks of two species, *Carcharhinus obscurus* and *Carcharhinus plumbeus*, presumably to spend the cold months at a higher temperature. This study concentrated on this point of artificial heat dissipation, which maintains a wide gradient of surface temperatures and allowed us to examine the temperature preferences of these species when given a larger range than what is naturally found in the sea. Between 2016 and 2018, 16 sharks were tagged with acoustic tags, 3 of which had temperature sensors, and 2 were additionally tagged with pop-up archival tags also logging temperature data. Results show that the sharks stayed in the elevated temperature, while the ambient sea was cold during the winter, spending several months in the heated area. Both species displayed a similar preferred range, spending 90 percent of their time at a temperature between 21.8 °C and 26.1 °C while the surrounding sea was 15.5–25.5 °C. Considering this chosen thermal niche and the rise in water temperature, it appears that for the past 40 years, the Eastern shores of the Mediterranean have become more suitable for these species, especially during transitional seasons. The question that arises, however, is whether these shark populations will benefit from the expanding range of preferable temperatures, or whether their proximity to shorelines will put them at greater risk in terms of human activities such as fishing and pollution.

**Keywords:** climate change; thermal niche; predators; range shifts; selacii; elasmobranch; habitat selection; *Carcharhinus obscurus*; *Carcharhinus plumbeus*; global warming

**Key Contribution:** Seasonal aggregations of Carcharhinid sharks are driven by a thermoregulatory behavior in which sharks remain within a specific range of temperatures. These findings provide valuable insights as to mechanisms that form these unique aggregations, and to further study the behavior and distribution of these species under global warning scenarios.





## 1. Introduction

Large coastal sharks are known to perform seasonal migrations for the purpose of feeding, reproduction, and thermoregulation. For example, for requiem sharks such as *Carcharhinus*

*falciformis*, movement patterns have been shown to differ in response to variations in resource abundance between distinct geographical regions [1]. In bull sharks (*C. leucas*), females have been reported to undergo seasonal migrations to give birth [2], and many other species are known to travel and migrate in sex-segregated cohorts (e.g., [3–5]). Migration in sharks can also constitute a mode of thermoregulatory behavior as they travel according to changes in ambient seawater temperatures while remaining within a specific temperature range (also referred to as a thermal niche (e.g., [6,7]). Understanding what drives and shapes migration patterns of large coastal sharks can prove essential for conservation efforts and predicting shark movements and distribution under predicted global changes in seawater temperatures.

*Carcharhinus obscurus* and *C. plumbeus* are among the requiem sharks (Carcharhinidae) with a cosmopolitan distribution that are also found in the Mediterranean Sea. Both species are large predators found in coastal and offshore waters [8–10] and are listed as endangered globally [11,12], with numbers of *C. plumbeus* reportedly declining by >70% over a period of 69 years [13].

For decades, large aggregations of carcharhinid sharks, comprised of *C. plumbeus* and *C. obscurus* occur every winter at Orot Rabin (OR) power station near Hadera, Israel in the Eastern Mediterranean and are not sighted at all during the rest of the year [14,15]. Arrival and departure of sharks at OR coincide with seasonal declines and elevations in seawater temperature, respectively. During the winter season, sharks at OR remain within a large plume of hot water discharge, which suggests that these aggregations are driven by a thermoregulatory behavior aimed at remaining within the sharks' thermal niche [14]. A similar seasonal pattern in the presence of *C. plumbeus* has also been reported in other parts of the Eastern Mediterranean—where sharks aggregate between May and August while sea water temperature ranges (20–28 °C).

Dusky sharks (*C. obscurus*) are rare in the Mediterranean Sea and are not observed aggregating anywhere aside from Israel [8,16,17]. The species was rarely encountered before the aggregations began, with less than 20 observations recorded [18], raising the question of whether their "new" appearance in the Mediterranean Sea is related to the possibility of spending the winter in a warm area. *C. plumbeus* is more common than *C. obscurus* in the Mediterranean Sea and is reported to aggregate in Bonçuk Bay, in Gökova Special Environmental Protection Area, Southwestern Turkey. A study from 2015 [19] found that sharks arrive at Bonçuk in spring and fall, during a temperature range of 20–27 °C.

While water temperature has been shown to correlate with the appearance of these aggregations [14,19], it remains unknown if individuals remain at OR for the duration of winter, as would be expected in the case of thermoregulatory behaviors, or if individuals remain for a far shorter time and are replaced by other individuals arriving throughout the season. The adjacent warm water discharge (10 °C above ambient sea temperature) also enabled us to empirically evaluate a thermal niche for the sharks at OR, as it allows individuals to easily control the temperature of their surroundings (i.e., by varying their distance to and from the outflow). Using acoustic telemetry and satellite tags equipped with temperature sensors we examined the preferred water temperature of individual sharks on site and examine what drives seasonal aggregations of sharks at OR.

## 2. Methods

### 2.1. Study Site

"Orot Rabin" (OR) station (32.466814N, 34.880232E) is a coal-fired power plant located near Hadera, Israel. OR has a long coal conveyor stretching two kilometres into the sea and uses six turbines to generate electricity. OR's turbines are cooled down by six double cooling systems pumping water from the sea. Water is used to cool down the turbines and is then discharged back to sea at up to 10 °C warmer than local conditions.

The resulting warm water plume forms a heated marine area along the coast a few kilometres south of OR and spreads approximately one to two kilometres west out to sea. The water temperature in the heated area is affected by the direction of currents, waves, winds, and turbine workload which is determined by temporary fluctuations in electricity

demand. This results in a study site where temperatures can change considerably from one day to the next.

### 2.2. Shark Tagging

Carcharhinid sharks at OR have been tagged since 2016 as part of an ongoing monitoring program conducted by the Morris Kahn marine research station, Israel (https://marsci.haifa. ac.il/en/the-morris-kahn-marine-research-station/ (accessed on 10 September 2022)). Shark movement in this study was monitored using acoustic transmitters and Pop-up Satellite Archival Tags (PSATs, MiniPAT-247A, Wildlife Computers, Redmond, WA, USA). A total of 16 sharks were tagged during 2 aggregation seasons: between January to February 2017 and between November 2017 to April 2018 (see Table 1 for details). All acoustic tags had a transmission interval of 30–90 s (nominal interval of 60 s) and operated at 69 kHz. The first acoustic receiver (VR2W, Vemco Inc., Bedford, NS, Canada) was deployed in OR's effluent on January 2017, and four additional receivers (TBR700, Thelma Biotel AS, Trondheim, Norway) were added in March 2018 (see Figure 1). PSATs were programmed to detach and transmit data after 192 days, to obtain a long tracking duration at a relatively high sampling frequency (every 60 s). PSATs were externally attached to the dorsal fin. Five of the sixteen sharks that were tagged also had sensors that provided in-situ measurements of ambient water temperature (see Table 1).

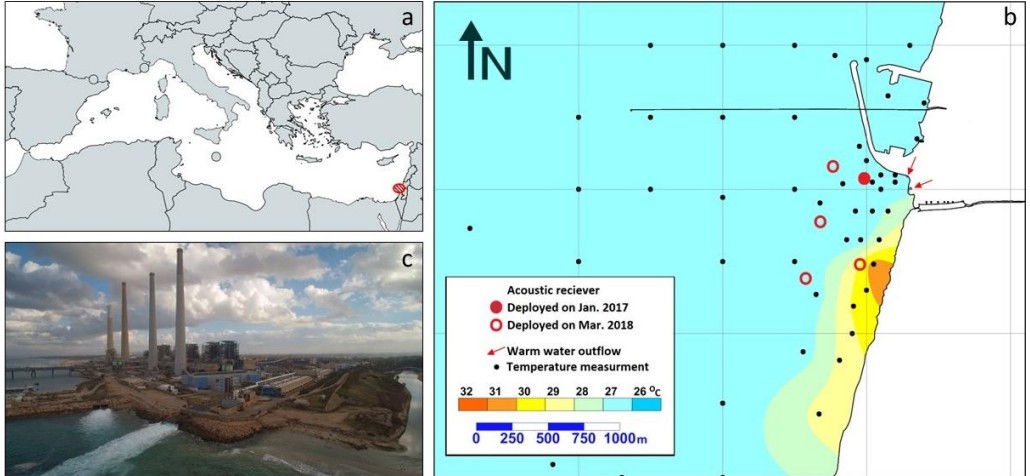

**Figure 1.** Study site map. (**a**) Study site location at the Easternmost end of the Mediterranean Sea. (**b**) Receivers' deployment at Orot Rabin power plant (OR) in Hadera. Temperature is shown as measured by IEC staff on 18 October 2018, at 2 m depth. Adapted from the IEC monitoring report 2018. (**c**) The warm water discharge at OR (Moshe Mittlman, January 2017).

Carcharhinid sharks, *C. obscurus*, and *C. plumbeus* were caught from a research vessel at OR's warm water effluent using baited lines. Once caught, sharks were pulled near the boat and strapped around the caudal peduncle and behind the pectoral fins to be measured and sexed. Sharks were then turned over and held in a state of tonic immobility, while acoustic tags were surgically implanted into the peritoneal cavity through a small incision. PSATs attached externally to the dorsal fin. Incisions were sutured immediately after the insertion of the acoustic transmitter, after which each tagged shark was released.

**Table 1.** Acoustic tagging study details. TL-total length. Temperature was measured by acoustic sensors for the first three individuals. Temperature ranges for individuals marked with asterisks were measured by archival tags tagged in addition to an acoustic transmitter with no temperature sensor. A nominal interval of 60 s.

| Shark Serial | Tagging Season | Species | Sex | TL (cm) | Transmitter Type (Sensors) | Detections | Tagging Date | Last Detected | Min Temp (°C) | Max Temp (°C) | Days Tracked | Detec./Day/Rec | Transmitter Model |
|---|---|---|---|---|---|---|---|---|---|---|---|---|---|
| 11941 | 2016–2017 | *C. obscurus* | F | 307 | Acoustic (temperature) | 1532 | 17 January 2017 | 12 April 2017 | 19.5 | 26.7 | 86 | 3.7 | V16T |
| 11942 | 2016–2017 | *C. obscurus* | F | 285 | Acoustic (temperature) | 1709 | 24 January 2017 | 18 April 2017 | 19.2 | 27.3 | 85 | 4.2 | V16T |
| 11943 | 2016–2017 | *C. obscurus* | F | 289 | Acoustic (temperature) | 242 | 28 February 2017 | 30 March 2017 | 19.5 | 24.5 | 31 | 0.6 | V16T |
| CO 21 | 2017–2018 | *C. obscurus* | F | 289 | acoustic | 318 | 27 November 2017 | 11 March 2018 | NA | NA | 105 | 3.2 | HP16 |
| CO 23 | 2017–2018 | *C. obscurus* | F | 276 | acoustic | 737 | 12 December 2017 | 24 April 2018 | NA | NA | 134 | 6.3 | HP16 |
| CO 22 | 2017–2018 | *C. obscurus* | F | 315 | acoustic | 482 | 27 December 2017 | 2 April 2018 | NA | NA | 97 | 6.1 | HP16 |
| CO 14 | 2017–2018 | *C. obscurus* | F | 355 | acoustic | 424 | 27 December 2017 | 13 March 2018 | NA | NA | 77 | 7.4 | HP16 |
| CO 20 | 2017–2018 | *C. obscurus* | F | 300 | acoustic | 267 | 2 January 2018 | 8 May 2018 | NA | NA | 127 | 3.1 | HP16 |
| CO 26 | 2017–2018 | *C. obscurus* | F | 275 | acoustic | 1051 | 5 February 2018 | 22 April 2018 | NA | NA | 77 | 8.2 | HP16 |
| CP 15 | 2017–2018 | *C. plumbeus* | M | 169 | acoustic | 17117 | 12 March 2018 | 14 May 2018 | NA | NA | 64 | 53.5 | HP16 |
| CO 25 | 2017–2018 | *C. obscurus* | F | 280 | acoustic | 63 | 12 March 2018 | 23 March 2018 | NA | NA | 12 | 1.1 | HP16 |
| CP 10 | 2017–2018 | *C. plumbeus* | M | 191 | acoustic | 17231 | 14 March 2018 | 10 May 2018 | NA | NA | 58 | 59.4 | HP16 |

**Table 1.** *Cont.*

| Shark Serial | Tagging Season | Species | Sex | TL (cm) | Transmitter Type (Sensors) | Detections | Tagging Date | Last Detected | Min Temp (°C) | Max Temp (°C) | Days Tracked | Detec./Day/Rec | Transmitter Model |
|---|---|---|---|---|---|---|---|---|---|---|---|---|---|
| CO 11 * | 2017–2018 | *C. obscurus* | F | 294 | Acoustic, PSAT | 969 | 28 March 2018 | 27 April 2018 | 22.3 * | 26.6 * | 31 | 6.3 | HP16 |
| CP 17 | 2017–2018 | *C. plumbeus* | M | 180 | acoustic | 4706 | 28 March 2018 | 14 May 2018 | NA | NA | 48 | 19.6 | HP16 |
| CO 12 | 2017–2018 | *C. obscurus* | F | 300 | acoustic | 1895 | 2 April 2018 | 2 June 2018 | NA | NA | 62 | 6.1 | HP16 |
| CP 27 * | 2017–2018 | *C. plumbeus* | M | 180 | Acoustic, PSAT | 4348 | 2 April 2018 | 21 April 2018 | 20.4 * | 26.8 * | 20 | 43.5 | HP16 |

*2.3. Water Temperature Measurements*

　　Water temperatures at OR are regularly measured by the Israel electric company (IEC) at the intake point of each pump and at the discharge point, also known as the warm water effluent. Water temperature measurements were supplied by the IEC environmental department and were measured every 30 min.

　　In this study, the median water temperature at the intake points represents ambient seawater temperatures close to shore whereas the temperature at the discharge point represents the maximum water temperature available at the site. The maximum temperature at the effluent fluctuated in conjunction with changes in pump operation so a median temperature of all functioning pumps was calculated for each 30-min time stamp, on both the intake and outtake data.

　　Individual temperature measurements were taken from two different tags. In the 2016–2017 season, three sharks were tagged internally with temperature sensors, providing data while the sharks were in the detection range of the receivers. In the 2017–2018 season, two sharks were successfully fitted with an external satellite tag, providing data regardless of the shark's location. The bottom depth near OR does not exceed 7.5 m, therefore data points from greater depths were removed for the temperature analysis.

*2.4. Mediterranean Water Temperature Measurements and Predictions*

　　Sea Surface Temperature (SST) measurements for the Mediterranean were downloaded from Copernicus Marine Service using the "SST MED SST L4 REP OBSERVATIONS 010 021" product which provides high-resolution optimally interpolated SST for the Mediterranean Sea on a daily (night-time) scale [20]. Temperature distribution maps were plotted for November, the month when sharks begin to aggregate at the power station (as previously reported in [14]). Three five-year periods were chosen to investigate thermal distribution in the Mediterranean: 1985–1990, 2000–2005, and 2015–2020 the latter corresponding to the time the data was collected in this study.

　　For future predictions, Representative Concentration Pathway (RCP) data were downloaded from the bio-oracle.org dataset [21] using RCP8.5, which is often used for predicting mid-21st century (and earlier) emissions based on current and stated policies [22]. Data for these scenarios are provided annually and not monthly, with the given options of maximum, minimum, or mean annual predictions. The mean annual temperature was chosen to best describe the temperatures in November (as a median between the lowest temperatures in February and the highest temperatures in August) in accordance with the maps of the previous time periods.

　　Thermal maps were created using Python Software [23]. The mean temperature was calculated for each time period but was only shown within the 90% quantiles of the temperature that sharks were found to inhabit in this study, to show the potential distribution area of *C. plumbeus and C. obscurus* on each map.

## 3. Results

*3.1. Residency and Date of Departure*

　　A total of 16 sharks (12 *C. obscurus* and 4 *C. plumbeus*) were caught at OR and tagged with internal acoustic tags. Out of the 16 tags three were equipped with temperature sensors (11941, 11942, and 11943). During the second tagging season (2017–2018) two sharks were tagged with Pop-up archival tags in addition to the acoustic tags. All caught *C. obscurus* sharks were females, ranging from 2.75 to 3.55 m and all caught *C. plumbeus* sharks were males ranging from 1.69 to 1.81 m, therefore, there was no overlap between species/sexes in size (Table 1). All sharks that were caught during the study were adult individuals.

　　Tracking duration after tagging ranged from 12 days up to 134 days. Individual sharks, especially the females *(C. obscurus)*, were found to spend months at the station (Table 1 and Figure 2) with a mean value for the tracked periods of 69.62 ± 9.02 SE days. We recognize

that our tracking period was limited due to tagging date in mid-season, therefore the actual time spent on site could be longer (Table 1).

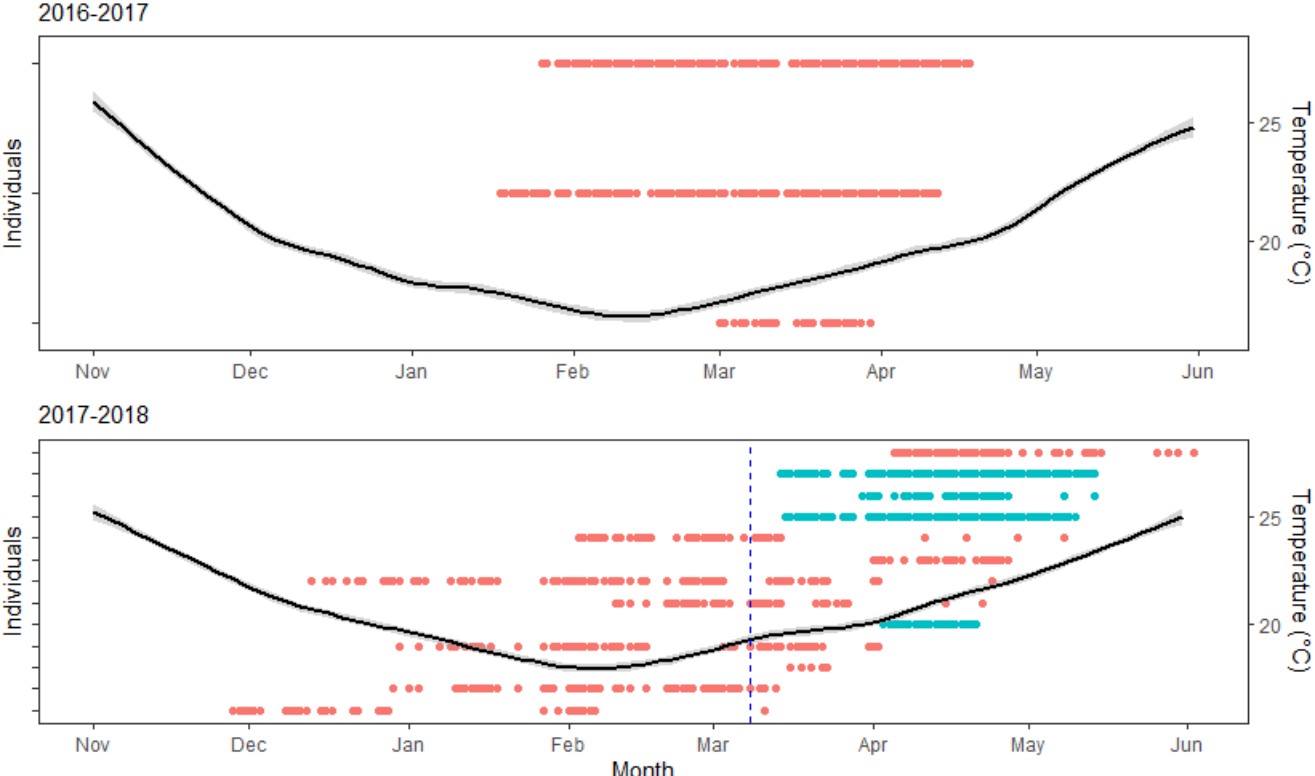

**Figure 2.** Shark detections and ambient sea water temperature. Shark detections are marked in blue for *C. plumbeus* and pink for *C. obscurus*. Smooth line of water temperature marked in black. The blue dashed line marks the addition of four receivers to the study site.

Individual CP 27 left the area shortly before the final exit and swam back in within a few hours. Depth and temperature measurements show a very clear separation between time spent in the heated area and out of it (Figure A1).

*3.2. Temperature*

Temperatures for individuals 11,941, 11,942, and 11,943 were extracted from the acoustic sensors, while temperature measurements for individuals 11 and 27 were extracted from the archival tags and corresponded with the periods these sharks also transmitted acoustically. Throughout the tagged period, sharks swam in a temperature between 19.16 °C and 27.32 °C and preferred swimming in the artificially elevated temperature, consistently keeping away from the ambient temperature (Figures 3 and A2). Sea temperature on time of departure (last detection) ranged from 18.62 °C to 24.91 °C, showing the same preferred range of temperature which individuals kept throughout the season (Figure 4). Temperatures on the day of departure were lower in the first season, probably due to having only one receiver, thus a smaller detection range. Individuals spent 90% of the time in a temperature between 21.8 °C and 26.05 °C and left the receivers' area only after the ambient temperature reached 19 °C and before the water temperature in the heated area reached 25 °C (Figures 2 and 3).

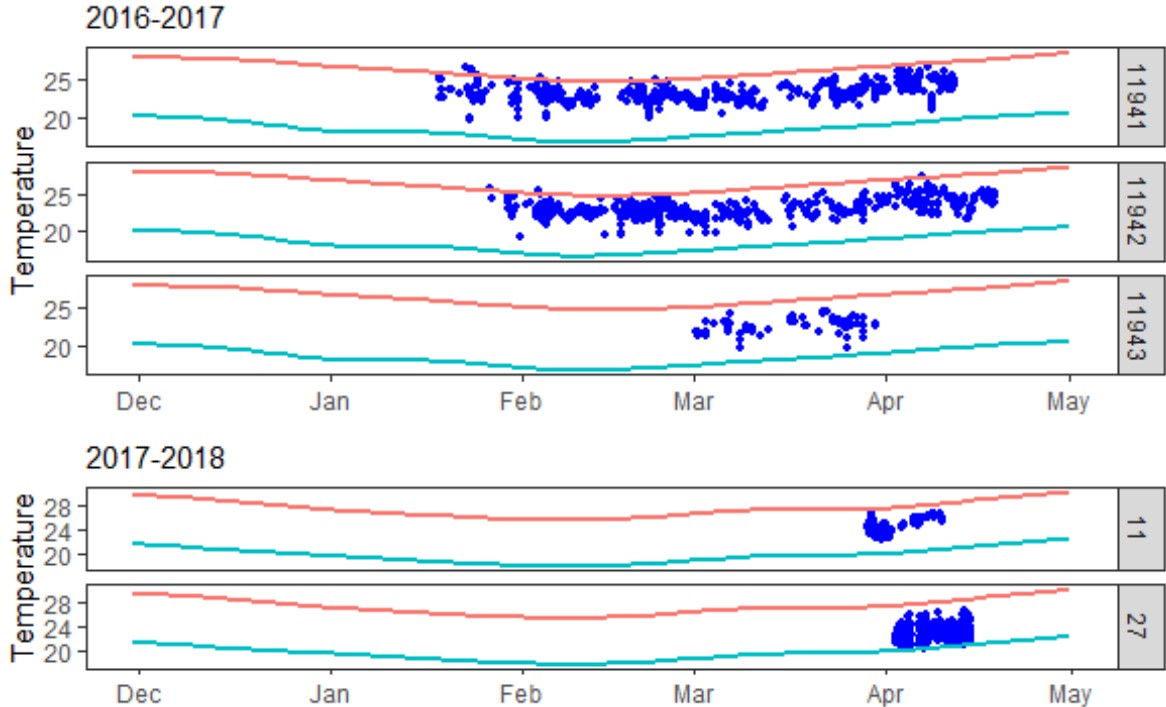

**Figure 3.** Temperature measurement for sharks within the warm effluent. Temperature measured by the shark tags shown in blue. Water temperature of the ambient sea is shown in the smooth light blue line and heated water temperature in the smooth red line.

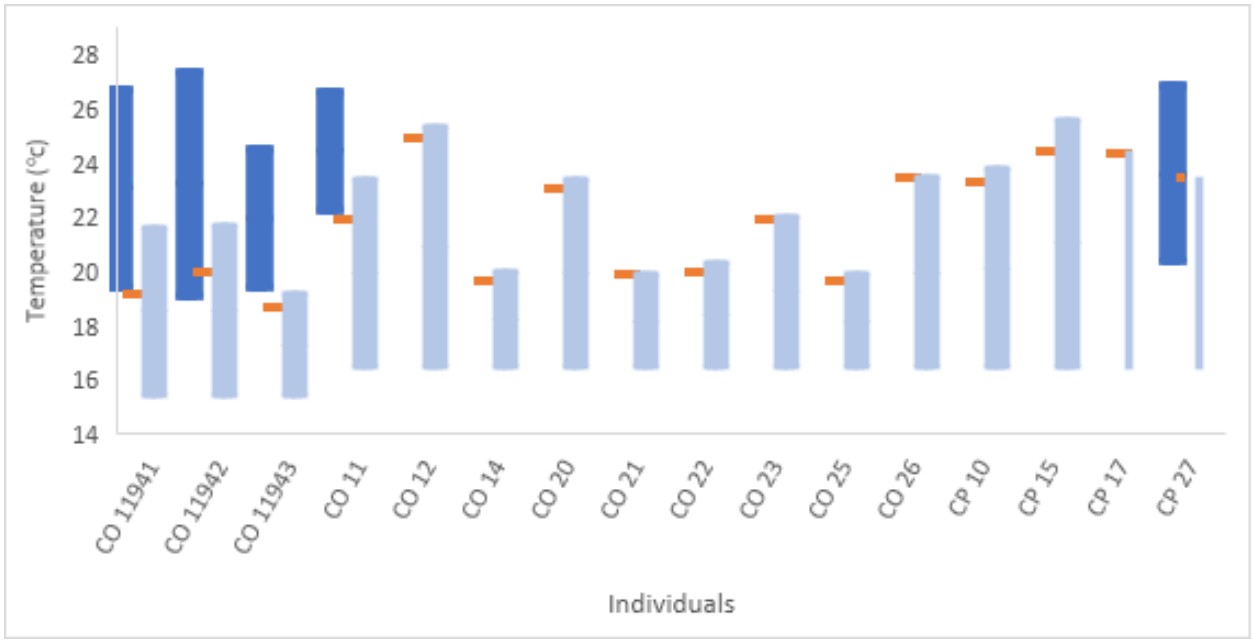

**Figure 4.** Water temperature (range) for each of the tagged sharks during their tracking time at OR, showing ambient sea water temperature, outside the plume of the warm water discharge (light blue), and temperature measured in-situ by the transmitters (dark blue). Orange markers represent the ambient sea temperature at the time of last detection (i.e., on leaving the study site). Sharks appear as CP for *C. plumbeus* and CO for *C. obscurus*. Note: The range of temperature measured from within the sharks is expectedly higher than ambient temperature as they mostly reside within the warm water discharge.

The only exception to this was shark 11,943 which was last detected in the area when the water temperature was 18.62 °C. This could be affected by having only one receiver in the water at that time, and therefore a reduced area was covered for detections.

During the second tagging season sharks appeared to be leaving the study area later in the season (Figure 2), when the ambient temperature is higher (Figure 4). The later time of last detection is probably also related to the added receivers and increased coverage of the array—which detected the tags while being farther away from the water discharge.

## 4. Discussion

This study provides the first mechanistic explanation for a seasonal aggregation of sharks at a coastal power plant in the Eastern Mediterranean. By use of in-situ temperature measurements of the sharks, we present evidence for sharks actively maintaining a thermal niche, and the possible role that interacting with the warm water discharge might play in the movements and migrations at a regional scale.

Sharks were observed at OR throughout the winter and all the individuals were detected at the study area at least until sea water temperature began to rise. A clear temperature range of 19–27 °C was observed to be maintained throughout the season, a range made possible only due to the heated water of the effluent. This now-recorded behavior reinforces the notion that thermoregulation is the underlying reason for shark aggregations at the site and explains the arrival and departure of sharks from the station as suggested in [14].

Similar conditions across individual arrival and departure (temperature), such as those observed at OR, have also presented in Bonçuk, Turkey [19], and are possibly indicative of migratory behavior at a regional scale (i.e., all arriving from somewhere at a particular time, and moving on close together in time). Sharks are known to migrate and aggregate in sex-segregated cohorts [4,5,24], and indeed we find within our data only females *C. obscurus* and only males *C. plumbeus*. This may reinforce the assumption that these aggregations are also related to reproduction. Gestation periods in these species stretch well over the time spent in the aggregation (approximately 2 years in *C. obscurus* [25])—therefore disentangling the two factors is problematic, and although the elevated temperature may benefit pregnant females, it is unlikely to be the sole driver, as males are also present at these aggregations. Higher prey densities or abundant resources cannot be eliminated as a factor in the sharks' attraction to the area. While anecdotal evidence from fishermen and divers suggests that teleost species may also be attracted by the hot water effluent, it remains undetermined whether prey availability is a contributing factor in attracting sharks to the area. Since sharks are strong swimmers and can easily come in and out of the area, our findings suggest this is unlikely as it would not explain the long residency of sharks, especially in light of substantially increased intra and interspecific competition. In cases where several species shared mutual feeding grounds, different species often tend to arrive at a specific time of the day, thus avoiding competition and high densities [26–28].

The power plant may facilitate or provide improved conditions for migrating sharks but also suggest a problematic dependency, especially given that these aggregations have been occurring for several decades [14]. A prolonged stay in an artificial location poses a significant effect on the life course of individuals within the population or even on the population as a whole. Spending extended periods in coastal waters in a highly urbanized area may subject sharks to sewage effluent, chemical pollution (e.g., heavy metals, pesticides), and noise pollution [29,30]. Research from Florida found juvenile nurse sharks (*Ginglymostoma cirratum*) exhibit lower levels of omega-6 highly unsaturated fatty acids and higher levels of both saturated and bacterial fatty acids as a result of proximity to urbanized areas [31]. Another study [32] from Florida suggests that the high numbers of infertility in bonnethead sharks *(Sphyrna tiburo)* in Tampa Bay may be linked to exposure to organochlorine contaminants.

Proximity to human activity exposes the sharks to uncontrolled tourism. The gathering of people at the power stations, swimmers, divers, and small vessels creates a permanent

disturbance to the natural behavior of sharks in the limited space they inhabit. The constant interaction between dozens of people and dozens of sharks can lead to unwanted interactions thus damaging the image of sharks and the public's willingness to protect them. The constant presence close to the shore in an accessible and well-known location also places sharks in danger of targeted fishing and bycatch of coastal fishing. Since all shark species are protected in Israel and fishing is prohibited, intentional fishing events have been rare in recent years. On the other hand, sharks are caught on a daily basis, and many are documented entangled with fishing hooks and other fishing gear [33–35].

In the marine environment, temperature plays a role in fish migratory movement and habitat selection [36–40]. Thermal niches for fish (defined as their preferred temperature $\pm 2$ °C or $\pm 5$ °C, Magnuson et al., 1979) differ among species [41], and sometimes among life stages [42,43] and/or sexes [44,45] within species.

Temperature was found to play a significant factor in triggering the emigration of juvenile *C. plumbeus* in South Carolina [46] and similar temperature preferences were found for the two species globally (Western Australia—[47], Hawaii—[48], and North Carolina—[49]). For *C. plumbeus*, most studies were investigating the movement behavior of juvenile sharks. While *C. plumbeus* sharks were found in a large range of temperatures, the majority of their time was spent in temperatures similar to those found in this study, or even higher (up to 30 °C).

Two of the tagged sharks in this study were equipped with PSATs which provide movement data beyond the study site. While one PSAT detached as soon as the shark left the study area near the power plant, the other individual was tracked for a duration of 10 days after exiting the acoustic array. Satellite tracking reveals that the shark travelled Southwest to the far side of Egypt in a directional trajectory and swam a total distance of 700 km until the shark appears to have been caught (mean distance of 70 km per day). The linear and persistent movement after leaving OR power plant may indicate that it intentionally aimed to reach a specific area or adhere to a particular route. Given the insufficient sample size and the scale of our study design, it remains unclear where sharks arrive from, or leave to, when they are not found at OR. Understanding the spatiotemporal context in which sharks aggregate at OR can provide clues as to the risks and benefits of such thermoregulatory behavior (e.g., [50]).

Several studies give evidence of the emerging effect of climate change and global warming on migratory species and observe changes in migratory patterns and seasonal distribution of terrestrial and avian animals due to changes in local temperatures worldwide (e.g., [51–55]). Other factors may potentially affect the occurrence and behavior of sharks at the power plant's effluent, such as salinity, energy conservation, and resource availability. The above factors are not easily disentangled from water temperature, and while the contribution of additional factors may indeed play a role, it is evident by our data that sharks persistently maintain a specific range of temperatures.

Marine environments are not spared from these rapid global changes (e.g., [55,56]), and the impact changing ocean temperatures have on marine ecosystems may be substantial [57,58], especially on ectothermic [59], k-selective, top predators such as sharks [60]. Recent research observed that rising sea temperatures have brought on changes in migratory timing and enabled some shark species to alter their distributional range [61–63].

Increasing temperatures in the Mediterranean Sea have been measured throughout the past four decades and are predicted to continue [20,64]. Changing sea temperatures can lead to significant differences in predator migration routes and consequently change the composition of entire ecosystems at a rapid rate [65]. Determining the preferred range of temperatures for these species is an important step in building estimation models for the expected distributions of the species in the future. Considering global warming and the high rate of seawater temperature rise in the Mediterranean Sea [66], these preferences could help predict changes in shark movement on a large scale.

With rising sea water temperature in the Mediterranean Sea, we found that the Eastern coast of the Mediterranean is becoming more accommodating for some carcharhinid sharks.

Between 1985 and 1990, only small areas in the East and South Mediterranean Sea exhibited the preferred temperature range for *C. obscurus* and *C. plumbeus*. Throughout the last 4 decades with the rise in SST, the compatible area has grown and stretches from Tunisia to West Turkey. The future scenario is predicting the preferred range will include almost the whole Eastern basin, including areas in Italy and Greece (Figure 5). It is possible that this temperature change might explain how these sharks "found" the stations and learned to use them during winter.

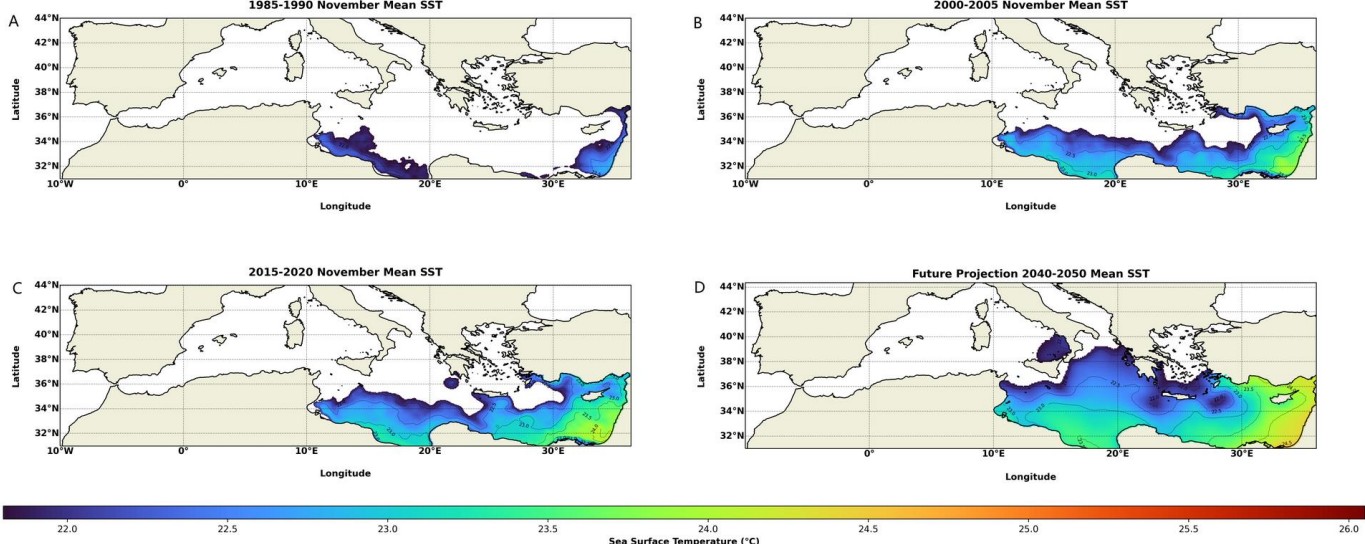

**Figure 5.** The mean surface temperature in the Mediterranean Sea, as measured by satellites for (**A**) 1985–1990 (**B**) 2000–2005 (**C**) 2015–2020 [20] and annual mean SST predictions by the RCP852050 scenario [67]. Temperatures of past years (panels **A**–**C**) are shown for the month of November when sharks begin to aggregate at the OR power station (as previously reported in [14]. Future annual predictions of SST (panel **D**) are represented as mean temperatures, to depict the expected rise in the month of November in the region. The range is shown only within the 90% quantile of temperature used by the sharks in this study (21.8 °C and 26.05 °C).

The understanding that sharks are migrating to an artificially heated area along with the relative speed at which sharks have learned to utilize the place and change their natural trajectory teaches us much in a time of changing environment and warming of sea temperature.

## 5. Conclusions

The timing of arrival and departure of adult Carcharhinid sharks at seasonal coastal aggregations may be dictated by sea water temperature, wherein sharks wait out the winter within a warm water discharge from a coastal power plant. These findings, though based only on a few individuals, may provide the first evidence of thermoregulatory behavior in sharks while undertaking seasonal migration, and clues as to mechanisms that underlie these unique aggregations. Temperature measurements from tagged sharks provide information on their thermal niche and how it is maintained. Finally, these findings are essential to better understand how rising sea temperatures in the Mediterranean Sea might affect sharks' migrations and distribution in the future.

**Author Contributions:** Conceptualization, A.B. and D.T.; Data curation, A.B., A.S., E.B., Z.Z.S. and S.M.; Formal analysis, A.B.; Investigation, A.B.; Methodology, A.B., R.P. and A.S.; Supervision, D.T.; Visualization, A.B. and Y.F.; Writing—original draft, A.B. and A.D.; Writing—review and editing, S.M., A.B., A.D. and R.P. All authors have read and agreed to the published version of the manuscript.

**Funding:** This study was funded by the Morris Kahn Marine Research Station, Department of Marine Biology, Leon H. Charney School of Marine Sciences, University of Haifa, Israel. This work was funded by the Kahn Foundation, and fieldwork was conducted by researchers of the Apex Marine Predator Laboratory of the Morris Kahn Marine Research Station.

**Institutional Review Board Statement:** Shark tagging was conducted under permit numbers 2017/41714 and 2018/42027, issued by The Israeli Nature and Parks Authority (INPA), and according to European ecological standards.

**Data Availability Statement:** Data are available from the authors upon reasonable request and with permission of the "Top Predator Lab" at Morris Kahn Marine Research Station, Department of Marine Biology, Leon H. Charney School of Marine Sciences, University of Haifa, Israel.

**Acknowledgments:** We thank Moshe Mittlman for his photograph (Figure 1), Kfir Avramzon the maritime lab manager in the engineering projects division, IEC, Israel for his help gathering and explaining IEC data, and Rotem Badash for her assistance with collecting and mapping temperature data.

**Conflicts of Interest:** The authors declare no conflict of interest.

## Appendix A

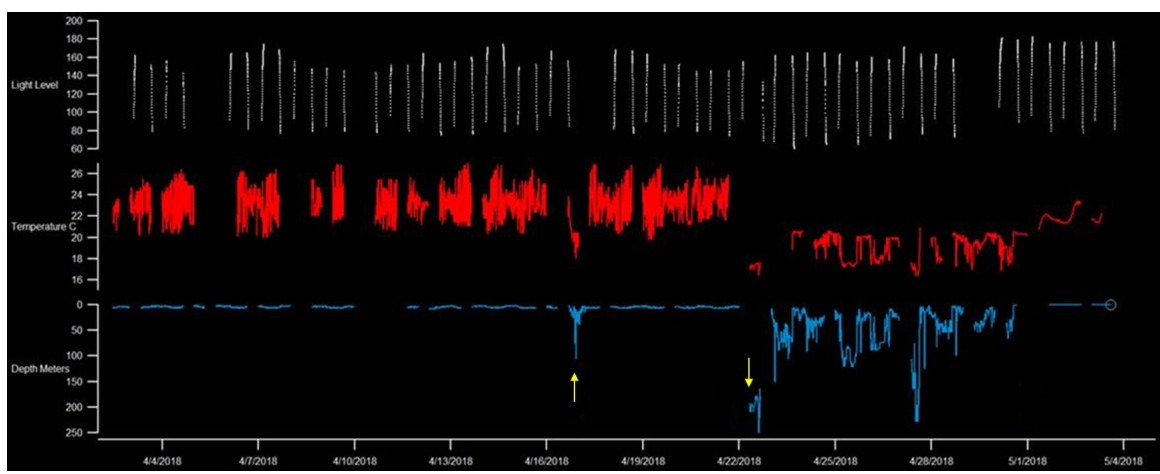

**Figure A1.** Light, temperature and depth records of *C. plumbeus* shark (ind. CP27) while being in the study area and at open sea. Yellow arrows indicate exits from the heated site.

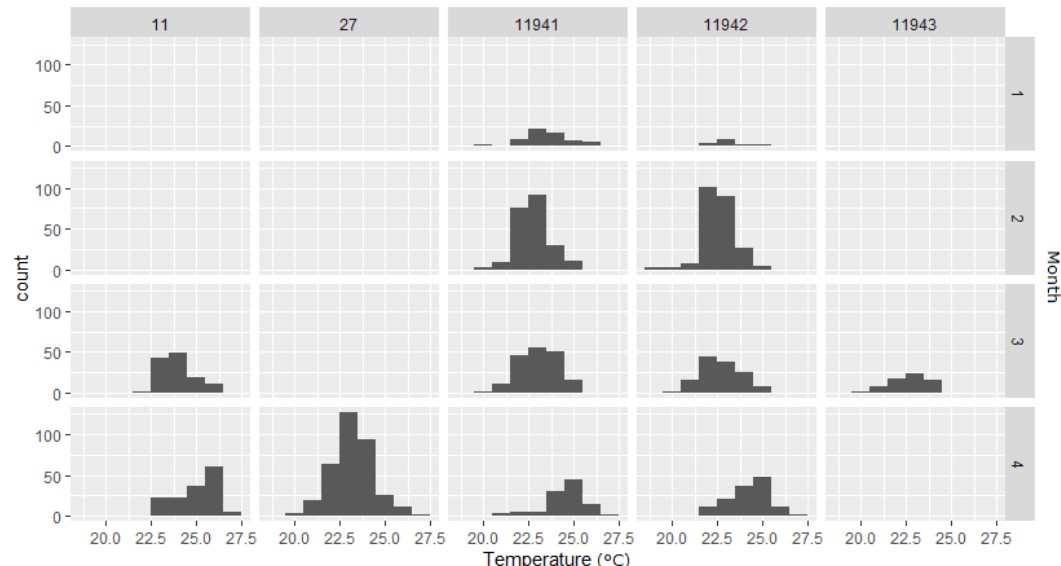

**Figure A2.** Temperature histogram describing detections (count) per temperature (mean temperature for 30 min) for each individual.

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
