# Peer review of "Some Like It Hot: Investigating Thermoregulatory Behavior of Carcharhinid Sharks in a Natural Environment with Artificially Elevated Temperatures"

_fishes, doi:10.3390/fishes8090428_

Round 1
Reviewer 1 Report
Please refer to comments provided in the attached file.

Reviewer 2 Report
Comments for Fishes 1954017
This MS describes the thermoregulatory behaviour of carcharhinid sharks in a natural environment with artificially elevated temperatures with acoustic and pop-up satellite archival tagging technique. This is an interesting topic and the topic fits the scope of the journal. However, the major problem of this MS is that it is not well organized, is lacking more insight descriptions from the tagging results, and the major conclusion of temperature preference is based on small sample size (n=5). The percentage of time spent in different temperature for tagged sharks should be presented. In addition, the vertical movement (depth) during day and night which may affect the temperature measured should be expressed separately. As the tagged sharks are all mature individuals, the interpretation can only be applied to adults. Why only large individuals were attracted by the warm water of the effluent and why these individuals gathering in this area should be explained and discussed. The authors claimed avoiding cold temperature is the reason but other possible environmental factors such as salinity or biological factors like feeding and reserving energy should be discussed. Discussion section needs to be intensively improved and many minor editorial errors need to be corrected.
In conclusion, I suggest the authors make a major revision by taking account my comments.
General comments
Does the warm water effluent of Orot Rabin also attract other teleost species such as the prey of the two shark species?
Usually, the abstract is one paragraph not separated to several paragraphs. I suggest the authors shorten the text and make it more concise.
L 48, in “marine fish” migratory movement.
L 68, Use Carcharhinus instead of C. for the first appearance.
L 86, I am not so sure that the results derived from this study can be applied to other regions. I suggest the authors delete “as well as world-wide”.
L 101, It is difficult to know the study location for those readers who are not familiar with this region. I suggest the authors add a map with larger scale (with latitude and longitude) and overlapping with the original one.
L 148, “2015-2020”. Why there are 10-year gaps in the three time periods?
L 157, Please clarify “therefor November)”?
L 171, The authors claimed that the tagging date was in mid-season and therefore the actual time spent on site could be longer. However, the detailed tagging date information was lacking in Materials and Methods section.
L 180, Fig. 2 is the pooled data for all individuals. As every individual was tagged in different date and tracking periods were different, it is difficult to understand individual behaviors such as moving in and out from this figure. The black line seems generated automatically without any model fitting.
L 181, Use Italian letter for C. plumbeus and C. obscurus.
L 185, “Fig. 2”.
L 194, “Figs. 2, 3”.
L 199, Please clarify that whether temperature showed in Fig. 3 of the two PSAT tagged individuals (no. 11 and 27) was in the same area as those from three acoustic tagged individuals.
L 200, “is shown the smoothen light”.
L 206, Use Italian letter for C. plumbeus and C. obscurus.
L 207-216, I suggest move this part including Fig. 5 to Discussion section.
The PSAT tagging method can track wide range of shark movement and residency, the results of PSAT should be used to verify or cross check with those from acoustic tagging experiments.
L 217, Discussion section is weak and needs to be improved.
L 224-233, In this study, only 18 individuals were caught and tagged. No other catch data support that adults of the two species will also stay in warm waters of the effluent.
L 236, “at least”.
L 238, “possible only due to the heated water of the effluent” is subjective. Other additional factors can not be ruled out.
L 253, “therefore”.
L 256, “Dusky sharks are rare …”.
L 261, Use Italian letter for C. plumbeus and C. obscurus.
L 265-273, As the two species are ecotherm not homeotherms as tunas, I am not sure that the movement of sharks will be directly affected by climate change. Indirect effect such as following their prey which are homeotherms to relocate the habitat is another possibility.
Round 2
Reviewer 1 Report
Please refer to attached PDF

Author Response
File with the corrections attached. thank you

Reviewer 2 Report
This revised MS has answered most of my general comments raised in the first round review but the authors did not response my major concern as followings: “The major problem of this MS is that it is not well organized, is lacking more insight descriptions from the tagging results, and the major conclusion of temperature preference is based on small sample size (n=5). The percentage of time spent in different temperature for tagged sharks should be presented. In addition, the vertical movement (depth) during day and night which may affect the temperature measured should be expressed separately. As the tagged sharks are all mature individuals, the interpretation can only be applied to adults. Why only large individuals were attracted by the warm water of the effluent and why these individuals gathering in this area should be explained and discussed. The authors claimed avoiding cold temperature is the reason but other possible environmental factors such as salinity or biological factors like feeding and reserving energy should be discussed.” Unless the authors can improve the above part, I will not recommend it for publishing.
Some minor editorial corrections are needed.
L 43, “Bull shark (C. leucas) females”.
L 53, “C. plumbeus”.
Figure 1, The notations of Deployed in Jan. 2017 and Deployed in March 2018 are the same. There seems something wrong.
Author Response
This revised MS has answered most of my general comments raised in the first round review but the authors did not response my major concern as followings: “The major problem of this MS is that it is not well organized, is lacking more insight descriptions from the tagging results, and the major conclusion of temperature preference is based on small sample size (n=5). The percentage of time spent in different temperature for tagged sharks should be presented. In addition, the vertical movement (depth) during day and night which may affect the temperature measured should be expressed separately. As the tagged sharks are all mature individuals, the interpretation can only be applied to adults. Why only large individuals were attracted by the warm water of the effluent and why these individuals gathering in this area should be explained and discussed. The authors claimed avoiding cold temperature is the reason but other possible environmental factors such as salinity or biological factors like feeding and reserving energy should be discussed.” Unless the authors can improve the above part, I will not recommend it for publishing.
We have, to the best of your understanding, make edits to the text in order to smooth line the reading and make the structure more coherent. To address your specific comments – the term adults added in Results, first paragraph, and conclusions.
in conclusion – inferences were toned down.
Additional/alternative factors are addressed explicitly and added to the discussion.
Some minor editorial corrections are needed.
L 43, “Bull shark (C. leucas) females”.
Corrected.
L 53, “C. plumbeus”.
Corrected.
Figure 1, The notations of Deployed in Jan. 2017 and Deployed in March 2018 are the same. There seems something wrong.
Thank you, corrected.
Round 3
Reviewer 2 Report
The revised MS has answered most of my questions raised in my previous review. I believe it can be accepted for publishing now.